

# Integer particle swarm optimization based task scheduling for device-edge-cloud cooperative computing to improve SLA satisfaction

Bo Wang[1], Junqiang Cheng[2], Jie Cao[1], Changhai Wang[1] and Wanwei Huang[1]

[1] Zhengzhou University of Light Industry, Zhengzhou, China
[2] Europe-Aisa Hi-tech and Digital Technology Company Limited, Zhengzhou, China

## ABSTRACT

Task scheduling helps to improve the resource efficiency and the user satisfaction for Device-Edge-Cloud Cooperative Computing (DE3C), by properly mapping requested tasks to hybrid device-edge-cloud resources. In this paper, we focused on the task scheduling problem for optimizing the Service-Level Agreement (SLA) satisfaction and the resource efficiency in DE3C environments. Existing works only focused on one or two of three sub-problems (offloading decision, task assignment and task ordering), leading to a sub-optimal solution. To address this issue, we first formulated the problem as a binary nonlinear programming, and proposed an integer particle swarm optimization method (IPSO) to solve the problem in a reasonable time. With integer coding of task assignment to computing cores, our proposed method exploited IPSO to jointly solve the problems of offloading decision and task assignment, and integrated earliest deadline first scheme into the IPSO to solve the task ordering problem for each core. Extensive experimental results showed that our method achieved upto 953% and 964% better performance than that of several classical and state-of-the-art task scheduling methods in SLA satisfaction and resource efficiency, respectively.

# INTRODUCTION

Smart devices, *e.g.,* Internet of Things (IoT) devices, smartphones, have been commonplace nowadays. "There were 8.8 billion global mobile devices and connections in 2018, which will grow to 13.1 billion by 2023 at a CAGR of 8 percent", as shown in the Cisco Annual Internet Report (*Cisco, 2020*). But most of the time, users' requirements cannot be satisfied by their respective devices. This is because user devices usually have limited capacity of both resource and energy (*Wu et al., 2019*). Device-Edge-Cloud Cooperative Computing (DE3C) (*Wang et al., 2020*) is one of the most promising ways to address the problem. DE3C extends the capacity of user devices by jointly exploiting the edge resources with low network latency and the cloud with abundant computing resources.

Task scheduling helps to improve the resource efficiency and satisfy user requirements in DE3C, by properly mapping requested tasks to hybrid device-edge-cloud resources. The

Corresponding author
Bo Wang, wangb@zzuli.edu.cn

goal of task scheduling is to decide whether each task is offloaded from user device to an edge or a cloud (offloading decision), which computing node an offloaded task is assigned to (task assignment), and the execution order of tasks in each computing node (task ordering) (*Wang et al., 2020*). To obtain a global optimal solution, these three decision problems must be concerned jointly when designing task scheduling. Unfortunately, to the best of our knowledge, there is no work that has jointly addressed all of these three decision problems. Therefore, in this paper, we try to address this issue for DE3C.

As the task scheduling problem is NP-Hard (*Du & Leung, 1989*), several works exploited heuristic methods (*Meng et al., 2019*; *Meng et al., 2020*; *Wang et al., 2019b*; *Yang et al., 2020*; *Liu et al., 2019*) and meta-heuristic algorithms, such as swarm intelligence (*Xie et al., 2019*; *Adhikari, Srirama & Amgoth, 2020*) and evolutionary algorithms (*Aburukba et al., 2020*; *Sun et al., 2018*), to solve the problem within a reasonable time. Inspired by a natural behaviour, a meta-heuristic algorithm has the capability of searching the optimal solution by combining a random optimization method and a generalized search strategy. meta-heuristic algorithms can have a better performance than heuristic methods, mainly due to their global search ability (*Houssein et al., 2021b*). Thus, in this paper, we design task scheduling for DE3C by exploiting Particle Swarm Optimization (PSO) which is one of the most representative meta-heuristics based on swarm intelligence, due to its ability to fast convergence and powerful ability of global optimization as well as its easy implementation (*Wang, Tan & Liu, 2018*).

In this paper, we focus on the task scheduling problem for DE3C to improve the satisfaction of Service Level Agreements (SLA) and the resource efficiency. The SLA satisfaction strongly affects the income and the reputation (*Serrano et al., 2016*; *Zhao et al., 2021*) and the resource efficiency can determine the cost at a large extent for service providers (*Gujarati et al., 2017*; *Wang et al., 2016*). We first present a formulation for the problem, and then propose a task scheduling method based on PSO with integer coding to solve the problem in a reasonable time complexity. In our proposed method, we encode the assignment of tasks to computing cores into the position of a particle, and exploit earliest deadline first (EDF) approach to decide the execution order of tasks assigned to one core. This encoding approach has two advantages: (1) it has much less solution space than binary encoding, and thus has more possibility of achieving optimal solution; (2) compared with encoding the assignment of tasks to computing servers (*Xie et al., 2019*; *Adhikari, Srirama & Amgoth, 2020*) (or a coarser resource granularity), our approach makes more use of the global searching ability of PSO. In addition, we use the modulus operation to restrict the range of the value in each particle position dimension. This can make boundary values have a same possibilities to other values for each particle position dimension, and thus maintain the diversity of particles. In brief, the contributions of this paper are as followings.

- We formulate the task scheduling problem in DE3C into a binary nonlinear programming with two objectives. The major objective is to maximize the SLA satisfaction, *i.e.,* the number of completed tasks. The second one is maximizing the resource utilization, one of the most common quantification approach for the resource efficiency.

- We propose an Integer PSO based task scheduling method (IPSO). The method exploits the integer coding of the joint solution of offloading decision and task assignment, and integrates EDF into IPSO to address the task ordering problem. Besides, the proposed method only restricts the range of particle positions by the modulus operation to maintain the particle diversity.
- We conduct simulated experiments where parameters are set referring to recent related works and the reality, to evaluate our proposed heuristic method. Experiment results show that IPSO has 24.8%–953% better performance than heuristic methods, binary PSO, and genetic algorithm (GA) which is a representative meta-heuristics based on evolution theory, in SLA satisfaction optimization.

In the rest of this paper, 'Problem Formulation' formulates the task offloading problem we concerned. 'IPSO Based Task Scheduling' presents the task scheduling approach based on IPSO. 'Performance Evaluation' evaluates scheduling approach presented in 'IPSO Based Task Scheduling' by simulated experiments. 'Discussion' discusses some findings from experimental results. 'Related Works' illustrates related works and 'Conclusion' concludes this paper.

## PROBLEM FORMULATION

In this paper, we focus on the DE3C environment, which is composed of the device tier, the edge tier, and the cloud tier, as shown in Fig. 1. In the device tier, a user launches one or more request tasks on its device, and processes these tasks locally if the device has available computing resources. Otherwise, it offloads tasks to an edge or a cloud. In the edge tier, there are multiple edge computing centres (edges for short). Each edge has network connections with several user devices and consists of a very few number of servers for processing some offloaded tasks. In the cloud tier, there are various types of cloud servers, usually in form of virtual machine (VM). The DE3C service provider, *i.e.*, a cloud user, can rent some instances with any of VM types for processing some offloaded tasks. The cloud usually has a poor network performance.

### Resource and task model

For our problem formulation, we consider the DE3C environment with $M$ user devices, $E$ edge servers, and $V$ cloud VMs. Without loss of generality, we use $n_i$, $i \in [1, M+E+V]$ to represent these computing nodes, where $n_i$ represent a device, an edge server, and a cloud VM respectively when $i \in [1, M]$, $i \in [M+1, M+E]$, and $i \in [M+E+1, M+E+V]$. The node $n_i$ has $C_i$ computing cores for processing tasks, and each core has $g_i$ computing capacity which can be quantified by such as Hz, FLOPS, or any other computing performance metric. The network bandwidth is $b_{i,j}$ ($i \in [1, M]$, $j \in [M+1, M+E+V]$) for transmitting data from device $n_i$ to node $n_j$, which can be easily calculated according to the transmission channel state data (*Du et al., 2019*). If a device is not covered by an edge, *i.e.*, there is no network connection between them, the corresponding bandwidth is set as 0.

In the DE3C environment, there are $T$ tasks, $t_1, t_2, \ldots, t_T$, requested by users for processing. We use binary constants $x_{o,i}, \forall o \in [1, T], \forall i \in [1, M]$, to represent the ownership

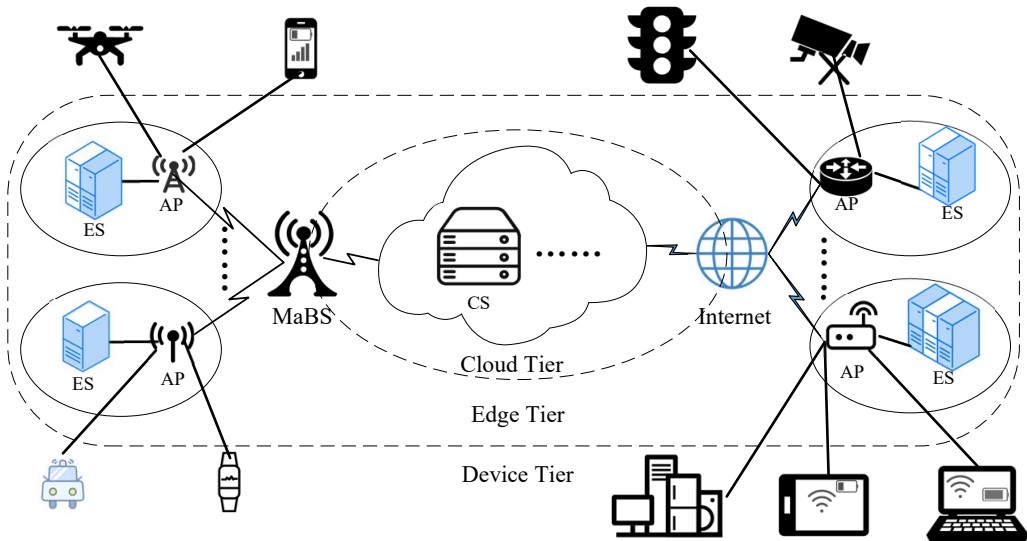

ES: Edge Server(s)   CS: Cloud Server(s)  AP: Access Point   MaBS: Macro Base Station

**Figure 1   The architecture of device-edge-cloud cooperative computing.**

relationships between tasks and devices, as defined in Eq. (1), which is known.

[1] In this paper, we focus on hard deadline tasks, and leave soft deadline tasks as a future consideration.

$$x_{o,i} = \begin{cases} 1, & \text{if } t_o \text{ is launched by } n_i \\ 0, & \text{else} \end{cases}, \forall o \in [1,T], \forall i \in [1,M]. \tag{1}$$

Task $t_o$ has $r_o$ computing length for processing its input data with size $a_o$, and requires that it must be finished within the deadline $d_o$[1]. Without loss of generality, we assume that $d_1 \le d_2 \le \dots \le d_T$. To make our approach universal applicable, we assume there is no relationship between the computing length and the input data size for each task.

## Task execution model

When $t_o$ is processed locally, there is no data transmission for the task, and thus, its execution time is

$$\tau_o = \sum_{i=1}^{M} (x_{o,i} \cdot \frac{r_o}{g_i}), \forall o \in [1,T]. \tag{2}$$

In this paper, we consider that each task exhausts only one core during its execution, as done in many published articles. This makes our approach more universal because it is applicable to the situation that each task can exhaust all resources of a computing node by seeing the node as a core. For tasks with elastic degree of parallelism, we recommend to referring our previous work (*Wang et al., 2019a*) which is complementary to this work.

Due to EDF scheme providing the optimal solution for SLA satisfaction maximization in each core (*Pinedo, 2016*), we can assume all tasks assigned to each core are processed in the ascending order of deadlines when establishing the optimization model. With this in

mind, the finish time of task $t_o$ when it processed locally can be calculated by

$$ft_o = \sum_{i=1}^{M} \sum_{q=1}^{C_i} (y_{o,i,q} \cdot \sum_{w=1}^{o} (y_{w,i,q} \cdot \tau_w)), \forall o \in [1, T], \tag{3}$$

where $y_{o,i,q}$ indicates whether task $t_o$ is assigned to the $q$th core of node $n_i$, which is defined in Eq. (4). $\sum_{w=1}^{o-1} (y_{w,i,q} \cdot \tau_w)$ is the accumulated sum of execution time of tasks which are assigned to $q$th core of $m_i$ and have earlier deadline than $t_o$, which is the start time of $t_o$ when it is assigned to the core. Thus, $\sum_{w=1}^{o} (y_{w,i,q} \cdot \tau_w)$ is the finish time when $t_o$ is assigned to $q$th core of $n_i$ ($i \in [1, M]$).

$$y_{o,i,q} = \begin{cases} 1, & \text{if } t_o \text{ is assigned to } q\text{th core of } n_i \\ 0, & \text{else} \end{cases}, o \in [1, T], i \in [1, M+E+V], q \in [1, C_i]. \tag{4}$$

As each task cannot be executed by any device which doesn't launch it,

$$\sum_{q=1}^{C_i} y_{o,i,q} \leq x_{o,i}, \forall o \in [1, T], \forall i \in [1, M]. \tag{5}$$

When a task is offloaded to the edge or the cloud tier, it starts to be executed only when both its input data and the core it is assigned are ready. Based on the EDF scheme, the ready time of input data for each task assigned to a core in an edge server or a VM can be calculated as

$$rt_o = \sum_{i=M+1}^{M+E+V} \sum_{q=1}^{C_i} (y_{o,i,q} \cdot \sum_{w=1}^{o} (y_{w,i,q} \cdot \frac{a_w}{b_{w,i}^T})), \forall o \in [1, T], \tag{6}$$

where $rt_o$ is the ready time of input data for task $t_o$ when the task is offloaded to an edge server or a cloud VM, respectively. For ease of our problem formulation, we use $b_{o,i}^T$ to respectively represent the network bandwidths of transferring the input data from the device launching $t_o$ to $n_i$ ($i \in [M+1, M+E+V]$). That is to say, $b_{o,i}^T = \sum_{k=1}^{M} (x_{o,k} \cdot b_{k,j})$. Then the transmission time of the input data for $t_o$ is $a_o / b_{o,i}^T$ when it is offloaded to $n_i$). With the EDF scheme, the ready time of the input data for an offloaded task is the accumulated transmission time of all input data of the offloaded task and other tasks which are assigned to the same core and have earlier deadline than the offloaded task, *i.e.*, $\sum_{w=1}^{o} (y_{w,i,q} \cdot a_w / b_{w,i}^T)$ for task $t_o$ when it is offloaded to $q$th core in $n_i$ ($i \in [M+1, M+E+V]$).

In this paper, we don't consider employing the task redundant execution for the performance improvement. Thus, each task can be executed by only one core, *i.e.*,

$$\sum_{i=1}^{M} \sum_{q=1}^{C_i} y_{o,i,q} \leq 1, \forall o \in [1, T]. \tag{7}$$

We use $z_o$ to indicate whether $t_o$ is assigned to a core for its execution, where $z_o = 1$ means yes and $z_o = 0$ means no. Then we have

$$z_o = \sum_{i=1}^{M} \sum_{q=1}^{C_i} y_{o,i,q}, \forall o \in [1, T]. \tag{8}$$

And the total number of tasks which are assigned to computing cores for executions is

$$Z = \sum_{o=1}^{T} z_o. \tag{9}$$

We use $ft_o^{off}$ to respectively represent the finish times of $t_o$ when it is offloaded to the edge or cloud tier. For $t_o$ assigned to a core, the core is available when all tasks that are assigned to the core and have earlier deadline than the task are finished. And thus, the ready time of the core for executing $t_o$ is the latest finish time of these tasks, which respectively are

$$rc_o = \sum_{i=M+1}^{M+E+V} \sum_{q=1}^{C_i} (y_{o,i,q} \cdot \max_{w<o} \{y_{w,i,q} \cdot ft_w^{off}\}), \forall o \in [1,T]. \tag{10}$$

The ready time of a task to be executed by the core it assigned to is the latter of the input data ready time and the core available time. The finish time of the task is its ready time plus its execution time. Thus, finish times of offloaded tasks are respectively

$$ft_o^{off} = \max\{rt_o, rc_o\} + \sum_{i=M+1}^{M+E+V} \sum_{q=1}^{C_i} (y_{w,i,q} \cdot \frac{r_o}{g_i}), \forall o \in [1,T]. \tag{11}$$

[2] In this paper, to avoid the negative effects on task execution performance, we don't consider to use dynamic frequency scaling technologies for computing energy saving.

Noticing that when a task is assigned to a tier, finish times of the task in other two tiers are both 0, as shown in Eqs. (3), and (11). Thus, the deadline constraints can be formulated as

$$ft_o + ft_o^{off} \leq d_o, \forall o \in [1,T]. \tag{12}$$

As the occupied time of each computing node is the latest usage time of its cores[2], and the usage time of a core is the latest finish time of tasks assigned to it. Therefore, the occupied times of computing nodes are respectively

$$ot_i = \max_{q \in [1,C_i]} \{ \max_{o \in [1,T]} \{y_{o,i,q} \cdot ft_o\}\}, \forall i \in [1,M], \tag{13}$$

$$ot_i = \max_{q \in [1,C_i]} \{ \max_{o \in [1,T]} \{y_{o,i,q} \cdot ft_o^{off}\}\}, \forall i \in [M+1,M+E+V]. \tag{14}$$

Then the total amount of occupied computing resources for task processing is

$$\Theta = \sum_{i=1}^{M+E+V} (ot_i \cdot C_i \cdot g_i). \tag{15}$$

And the overall computing resource utilization of the DE3C system is

$$U = \frac{\sum_{o=1}^{T}(y_o \cdot r_o)}{\Theta}, \tag{16}$$

where the numerator is the accumulated computing length of executed tasks, *i.e.,* the amount of computing resource consumed for the task execution.

## Problem model

Based on above formulations, we can model the task scheduling problem for DE3C as

**Maximizing** $Z + U$            (17)

    subject to

$(2) - (16),$            (18)

where the objective Eq. (17) is maximizing the number of finished tasks ($Z$), which is considered as the quantifiable indicator of the SLA satisfaction in this paper, and maximizing the overall computing resource utilization ($U$) when the finished task number cannot be improved (noticing that the resource utilization is no more than 1). The decision variables include $y_{o,i,q}$ ($q \in [1, C_i]$, $i \in [1, M + E + V]$, $o \in [1, T]$). This problem is binary nonlinear programming (BNLP), which can be solved by existing tools, *e.g.,* lp_solve (*Berkelaar et al., 2020*). But these tools are not applicable to large-scale problems, as they are implemented based on branch and bound. Therefore, we propose a task scheduling method based on an integer PSO algorithm to solve the problem in a reasonable time in the next section.

## IPSO BASED TASK SCHEDULING

In this section, we present our integer PSO algorithm (IPSO) based task scheduling method in DE3C environments to improve the SLA satisfaction and the resource efficiency. Our proposed method, outlined in Algorithm 1, first employs IPSO to achieve the particle position providing the global best fitness value in Algorithm 3, where the position of each particle is the code of the assignment of tasks to cores. Then our IPSO based method can provide a task scheduling solution according to the task assignment get from the previous step by exploiting EDF scheme for task ordering in each core, as shown in Algorithm 2. In our IPSO, to quantify the quality of particles, we define the fitness function as the objective (Eq. 17) of the problem we concerned,

$fn = Z + U,$            (19)

where $Z$ is the number of finished tasks, and $U$ is the overall computing resource utilization. In the followings, we will present the integer encoding and decoding approach exploited by the IPSO in 'Integer Encoding and Decoding', and the detail of IPSO in 'IPSO'.

---

**Algorithm 1** IPSO based task scheduling

---

**Input:** The information of tasks and resources in the DE3C system; the integer encoding and decoding method.
**Output:** A task scheduling solution.
  1: achieving the global best particle position by IPSO (see Algorithm 3);
  2: decoding the position into a task scheduling solution by Algorithm 2;
  3: **return** the task scheduling solution;

---

## Integer encoding and decoding

Our IPSO exploits the integer encoding method to convert a task assignment to cores into the position information of a particle. We first respectively assign sequence numbers to tasks

---

**Algorithm 2** Converting a particle position to a task scheduling solution

---

**Input:** The particle position, $[gb_1, gb_2, \ldots, gb_T]$.
**Output:** The task scheduling solution and the fitness.
  1: decoding the position to the assignment of tasks to computing cores;
  2: **for** each core **do**
  3:     sorting the task execution order in the descending on the deadline;
  4: calculating the fitness using Eq. (19);
  5: **return** the task scheduling solution and the fitness;

---

and cores, both starting from one, where the task number is corresponding to a dimension of a particle position, and the value of a dimension of a particle position is corresponding to the core the corresponding task assigned to. For cloud VMs, we only number one core for each VM type as all VM instances with a type have identical price-performance ratio in real world, *e.g.,* a1.\* in Amazon EC2 (https://aws.amazon.com/). For each task, the number of cores which it can be assigned to is the accumulated core number of the device launching it and the edge servers having connections with the device plus the number of VM types. Thus, for each dimension in each particle position, the minimal value ($p_d^{min}$) is one representing the task assigned to the first core of the device launching the task, and the maximal value ($p_d^{max}$) is

$$p_d^{max} = \sum_{i=1}^{M}(x_{d,i} \cdot C_i) + \sum_{i=1}^{M}\sum_{j=M+1}^{M+E}\sum_{b_{i,j}>0}(x_{d,i} \cdot C_j) + NV, \tag{20}$$

where $NV$ is the number of cloud VM types. The subscript $d$ represents the dimension in each particle position. The $d$th dimension is corresponding to the $d$th task, $t_d$.

For example, as shown in Fig. 2, assuming a DE3C consisting of two user devices, one edge server, and one cloud VM type. Each of these two devices and the edge server has two computing cores, respectively represented as $dc_{11}$ and $dc_{12}$ for the first device, $dc_{21}$ and $dc_{22}$ for the second device, and $ec_1$ and $ec_2$ for the edge server. Both devices have network connections with the edge, that is to say, tasks launched by these two devices can be offloaded to the edge for processing. Then for each task, there are five cores that the task can be assigned to, *i.e.*, $p_d^{max} = 5$ for all $d$. Each device launches three tasks, where the first three tasks, $t_1$, $t_2$, and $t_3$, are launched by the first device, and the last three tasks, $t_4$, $t_5$, and $t_6$, are launched by the second device. Then we numbered two cores of each device as 1 and 2, respectively. Two cores of the edge server are numbered as 3 and 4, respectively. And the VM type is numbered as 5. By this time, the particle position $[2, 3, 2, 1, 4, 5]$ represents $t_1$ is assigned to $dc_{12}$, $t_2$ is assigned to $ec_1$, $t_3$ is assigned to $dc_{12}$, $t_4$ is assigned to $dc_{21}$, $t_5$ is assigned to $ec_2$, and $t_6$ is assigned to the cloud.

Given a particle position, we use the following steps to convert it to a task scheduling solution, as outlined in Algorithm 2: (i) we decode the position to the task assignment to cores based on the correspondence between them, illustrated above; (ii) with the task assignment, we conduct EDF for ordering the task execution on each core, which rejects tasks whose deadline cannot be satisfied by the core, as shown in lines 2–3 of Algorithm 2. By now, we achieve a task scheduling solution according to the particle position. After this, we can calculate the number of tasks whose deadlines are satisfied, and the overall

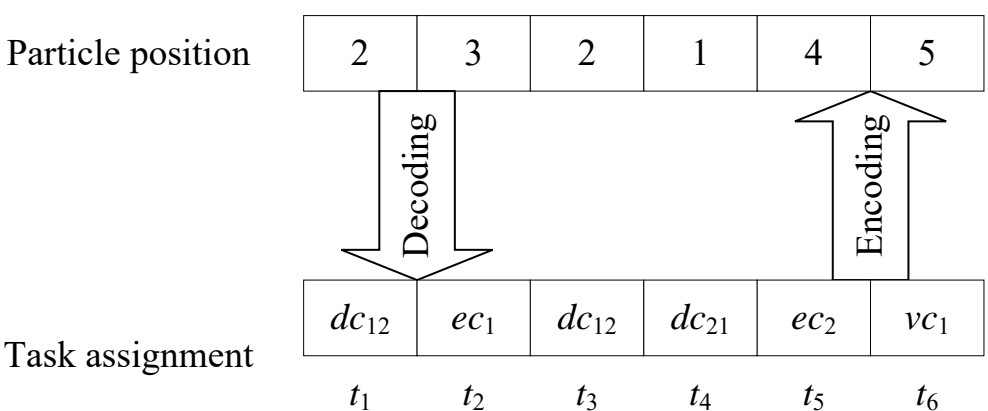

**Figure 2** An example for illustrating the integer coding method.

utilization using Eqs. (13)–(16). And then, we can achieve the fitness of the particle using Eq. (19).

**IPSO**

The IPSO, exploited by our method to achieve the best particle position providing the global best fitness, consists of the following steps, as shown in Algorithm 3.

1) Initializing the position and the fly velocity for each particle randomly, and calculating its fitness.

2) Setting the local best position and fitness as the current position and fitness for each particle, respectively.

3) Finding the particle providing the best fitness, and setting the global best position and fitness as its position and fitness.

4) If the number of iterations don't reach the maximum predefined, repeat step (5)–(7) for each particle.

5) For the particle, updating its velocity and position respectively using Eqs. (21) and (22), and calculating its fitness. Where $v_{i,d}$ and $p_{i,d}$ are representing the velocity and the position of $i$th particle in $d$th dimension. $lb_{i,d}$ is the local best position for $i$th particle in $d$th dimension. $gb_d$ isthe global best position in $d$th dimension. $\omega$ is the inertia weight of particles. We exploit linearly decreasing inertia weight in this paper, due to its simplicity and good performance (*Han et al., 2010*). $a_1$ and $a_2$ are the acceleration coefficients, which push the particle toward local and global best positions, respectively. $r_1$ and $r_2$ are two random values in the range of $[0, 1]$. To rationalize the updated position in $d$th dimension, we perform rounding operation and modulo $p_i^{max}$ plus 1 on it.

$$v_{i,d} = \omega \cdot v_{i,d} + a_1 \cdot r_1 \cdot (lb_{i,d} - v_{i,d}) + a_2 \cdot r_2 \cdot (gb_d - v_{i,d}), \tag{21}$$

$$p_{i,d} = \lceil p_{i,d} + v_{i,d} \rceil \bmod p_i^{max} + 1. \tag{22}$$

The inertia weight update strategy and the values of various parameters (*e.g.*, $a_1$ and $a_2$) have influences on the performance of PSO, which is one of our future works. One is advised to read related latest works, *e.g.*, (*Houssein et al., 2021a*; *Nabi & Ahmed, 2021*), if interested to follow the details.

6) For the particle, comparing its current fitness and local best fitness, and updating the local best fitness and position respectively as the greater one and the corresponding position.

7) Comparing the local best fitness with the global best fitness, and updating the global best fitness and position respectively as the greater one and the corresponding position.

---

**Algorithm 3** IPSO

---

**Input:**   the parameters of the IPSO.
**Output:**   The global best particle position.
  1:  generating the position and the velocity of each particle randomly;
  2:  calculating the fitness of each particle by Algorithm 2;
  3:  initializing the local best solution as the current position and the fitness for each particle;
  4:  initializing the global best solution as the local best solution of the particle providing the best fitness;
  5:  **while** the iterative number don't reach the maximum **do**
  6:     **for** each particle **do**
  7:        updating the position using Eq. (21) and (22);
  8:        calculating the fitness of each particle by Algorithm 2;
  9:        updating the local best solution;
10:        updating the global best solution;
11:  **return**  the global best solution;

---

For updating particle positions, we only discretize them and limit them to reasonable space, which is helpful for preserving the diversity of particles. Existing discrete PSO methods limit both positions and velocity for particles, and exploit the interception operator for the limiting, which sets a value as the minimum and the maximum when it is less than the minimum and greater than the maximum, respectively. This makes the possibilities of the minimum and the maximum for particle positions are much greater than that of other possible values, and thus reduces the particle diversity, which can reduce the performance of PSO.

## Complexity analysis

As shown in Algorithm 3, there are two layers loop, which has $O(ITR \cdot NP)$ time complexity, where $ITR$ and $NP$ are the numbers of the iteration and particles, respectively. Within the loop, the most complicated part is calling Algorithm 2 which is $O(NC \cdot (T/NC)^2) = O(T^2/NC)$ in time complexity on average, as shown in its lines 2–3, where $NC$ is the number of cores in the DE3C system. $T/NC$ is the average number of tasks assigned to each core, and $O((T/NC)^2)$ is the time complexity of EDF for each core on average. Thus, the time complexity of our IPSO based method is $O(ITR \cdot NP \cdot T^2/NC)$ on average, which is quadratically increased with the number of tasks.

For PSO or GA with binary encoding method, they have similar procedures to IPSO, and thus their time complexities are also $O(ITR \cdot NP \cdot T^2/NC)$. Referring to *Bays (1977)*; *B.V. & Guddeti (2018)*; *Benoit, Elghazi & Robert (2021)*; *Liu et al. (2019)*; *Meng et al. (2020)*; *Mahmud et al. (2020)*, time complexity of First Fit (FF) is $O(T * NC)$, and that of First Fit Decreasing (FFD), Earliest Deadline First (EDF), Earliest Finish Time First (EFTF), Least Average Completion Time (LACT), and Least Slack Time First (LSTF) are $O(T^2 * NC)$. In general, the numbers of the iteration and particles are constants, and all of above methods except FF exhibit quadratic complexity with the number of tasks.

---

# PERFORMANCE EVALUATION

In this section, we conduct extensive experiments in a DE3C environment simulated referring to published articles and the reality, to evaluate the performance of our method in SLA satisfaction and resource efficiency. In the simulated environment, there are 20 devices, 2 edges, and one cloud VM type, and each device is randomly connected with one edge. For each edge, the number of servers are randomly generated in the range of $[1, 4]$. The computing capacity of each core is randomly set in the ranges of $[1, 2]$ GHz, $[2, 3]$ GHz, and $[2, 3]$ GHz, respectively, for each device, each edge server, and the VM type. The number of tasks launched by each device is generated randomly in the range of $[1, 100]$, which results in about 1,000 total tasks on average in the system. The length, the input data size, and the deadline of each task is generated randomly in the ranges of $[100, 2000]$ GHz, $[20, 500]$ MB and $[100, 1000]$ seconds, respectively. For network connections, the bandwidths for transmitting data from a device to an edge and the cloud are randomly set in ranges of $[10, 100]$ Mbps and $[1, 10]$ Mbps, respectively.

There are several parameters should be set when implementing our IPSO. Referring to *Kumar, Mahato & Bhunia (2020)*, we set the maximum iteration number and the particle number as 200 and 50, respectively. Both acceleration coefficients, $a_1$ and $a_2$, are set as 2.0, referring to *Wang, Zhang & Zhou (2021)*. The $\omega$ inertia weight is linearly decreased with the iterative time in the range of $[0.0, 1.4]$, referring to *Yu et al. (2021)*. The effect of parameter settings on the performance of our method will be studied in the future.

We compare our method with the following classical and recently published methods.

- First Fit (*Bays, 1977*), FF, iteratively schedules a task to the first computing core satisfying its deadline.
- First Fit Decreasing (*B.V. & Guddeti, 2018*), FFD, iteratively schedules the task with maximal computing length to the first computing core satisfying its deadline.
- Earliest Deadline First (*Benoit, Elghazi & Robert, 2021*), EDF, iteratively schedules the task with earliest deadline to the first computing core satisfying its deadline, which is a classical heuristic method concerning the deadline constraint.
- Earliest Finish Time First, EFTF, iteratively schedules a task to the computing core providing the earliest finish time and satisfying its deadline, which is the basic idea exploited in the work proposed by *Liu et al. (2019)*.
- Least Average Completion Time, LACT, iteratively schedules a task to the computing core satisfying its deadline, such that the average completion time of scheduled tasks is minimal, which is the basic idea exploited by Dedas (*Meng et al., 2020*).
- Least Slack Time First, LSTF, iteratively schedules a task to the computing core satisfying its deadline and providing the least slack time for the task, which is the basic idea exploited in the work proposed by *Mahmud et al. (2020)*.
- Genetic Algorithm, GA, which simulates the population evolution by crossover, mutation and selection operators, where a chromosome represents a task assignment to cores, and a gene is a bit representing the assignment of a task to a core. This is the basic idea used in the work of *Aburukba et al. (2020)*. The number of population and the

maximum generation are set as 1,000 in our experiments. EDF is used for task ordering in each core.

- PSO_srv, employ PSO with the integer coding of the task assignment to computing nodes, which is the basic idea exploited by existing PSO based task scheduling for DE3C, such as *Xie et al. (2019)*. Parameters have same settings to our IPSO method in the experiment. The assignment of tasks to cores in each computing node and the task ordering in each core are solved by EDF.
- BPSO, employ PSO with the binary coding that is similar to GA. Parameter values are set as our IPSO method in the experiment. EDF is used to order the task execution in each core.

We compare the performance of these task scheduling method in the following aspects.

- SLA satisfaction is quantified by the number, the accumulated computing length, and the processed data size of completed tasks.
- Resource efficiency is quantified by the resource utilization for the overall system, and the cost efficiency for cloud VMs. The price of a VM instance is $0.1 per hour.
- Processing efficiency is quantified by the executed computing length and the processed data size of tasks per unit processing time, which are the ratios of completed computing length and processed data size to the makespan, respectively.

For each group of experiments, we repeat it ten times, and report the median result in the followings. For each metric value achieved by each task scheduling method, we scale it by that of FF to display the relative difference between these methods more clearly. The details of experiment results are shown as followings. Our method is abbreviated to IPSO in the followings.

### SLA satisfaction

Figure 3 shows the performance of various task scheduling methods in maximizing the accumulated number, computing length, and processed data size of finished tasks. As shown in the figure, our method has about 36%, 46%, and 55% better performance than these heuristic methods, FF, FFD, EDF, EFTF, LACT, and LSTF, in these three SLA satisfaction metrics, respectively. This illustrates that meta-heuristic methods can achieve a much better performance than heuristic methods, due to their randomness for global searching ability. While, the performance of GA, PSO_srv, and BPSO are much worse than that of these heuristic methods, such as, our method achieves 953%, 115%, and 322% than GA, PSO_srv, and BPSO, respectively, more completed task number. This suggests us that meta-heuristic approaches must be carefully designed for a good performance.

Of these heuristic methods, EDF has the best performance in SLA satisfaction, as shown in Fig. 3, due to its aware of the task deadline, which is the reason why we exploit it for task ordering in each core.

Compared with BPSO, IPSO has 322%, 523%, and 313% better performance in three SLA satisfaction metrics, respectively. This provides experimental evidence that our integer coding method significantly improves the performance of PSO for task scheduling in DE3C. The main reason is that the searching space BPSO is much larger than IPSO due to

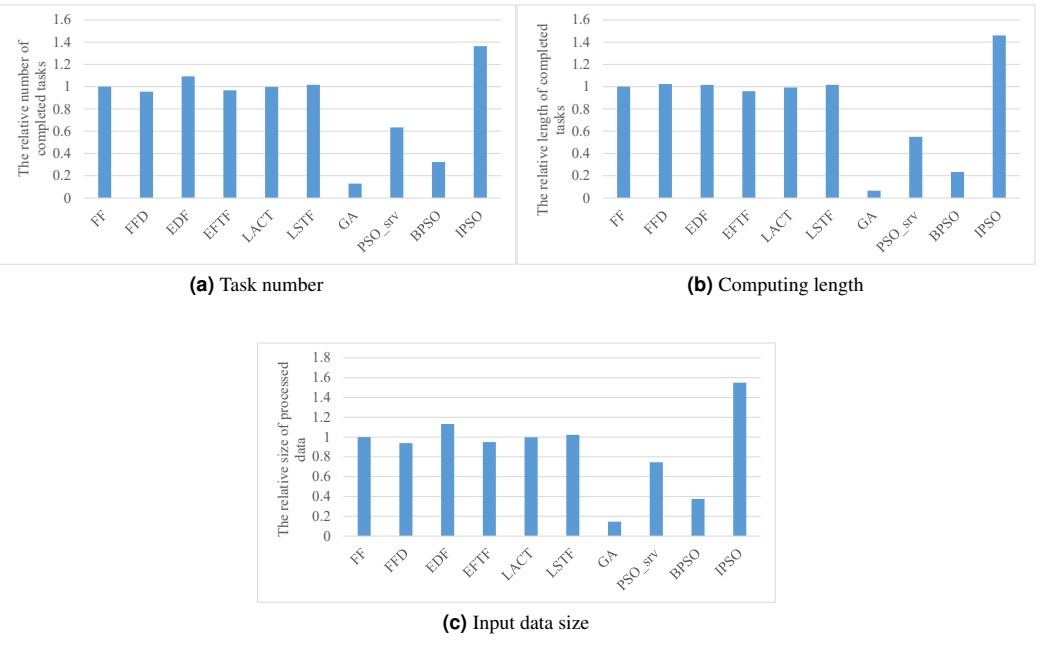

**(a)** Task number      **(b)** Computing length

**(c)** Input data size

**Figure 3** **The SLA satisfaction performance achieved by various scheduling methods.** (A) Task number, (B) Computing length (C) Input data size.

their different represents to the same problem. For example, if there are 6 candidate cores, *e.g.,* two device cores, four edge server cores, and one cloud VM type, for processing every task in a DE3C, then considering there are 10 tasks, the search space includes $6^{10}$ solutions for IPSO, while $2^{60}$ for BPSO. Thus, in this case, the search space of BPSO is more than 330 million times larger than that of IPSO, and this multiple exponentially increases with the numbers of tasks and candidate cores of each task. Therefore, for an optimization problem, IPSO has much probability of searching a local or global best solution than BPSO. This is also the main reason why GA has much worse performance than IPSO, as shown in Fig. 3.

PSO_srv has smaller search space while worse performance than IPSO. As shown in Fig. 3, IPSO has 115%, 166%, and 108% better performance than PSO_srv in three SLA satisfaction metrics, respectively. This is mainly because the coding of the task assignment to a coarse-grained resource cannot take full advantage of the global searching ability of PSO. Google's previous work has verified that fine-grained resource allocation helps to improve the resource efficiency (*Tirmazi et al., 2020*), thus our IPSO uses the core as the granularity of resources during the searching process, which results in a better performance in SLA satisfaction optimization compared with other methods.

## Resource efficiency

Figures 4A and 4B respectively show the overall resource utilization and the cost efficiency when using various task scheduling methods in the simulated DE3C environment. As shown in the figure, our method has the best performance in optimizing both resource efficiency metric values, where our method has 36.9%–964% higher resource utilization

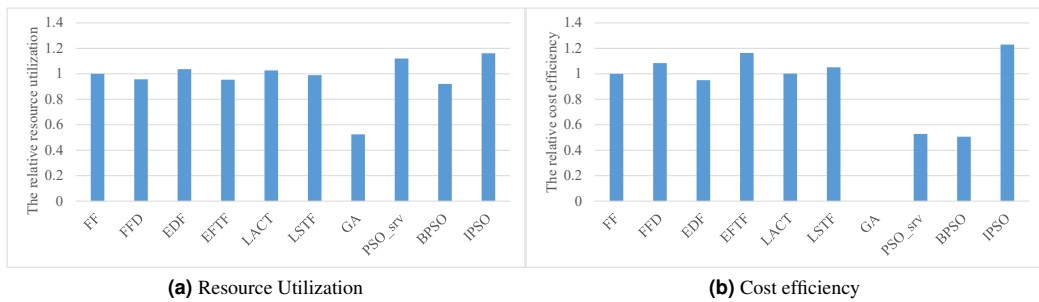

**(a)** Resource Utilization                **(b)** Cost efficiency

**Figure 4** **The resource efficiency achieved by various scheduling methods.** (A) Resource utilization, (B) Cost efficiency.

and 5.6%–143% greater cost efficiency, compared with other methods (GA has zero cost efficiency as its solution does not offload any task to the cloud). This is mainly because the resource utilization is our second optimization objective (see Eqs. (17) or (19)), which results in that the solution having a higher utilization has a greater fitness in all of the solution with same number of finished tasks.

EDF has a worse cost efficiency than other heuristic methods. This may be because EDF offloads more tasks to the cloud, which leads to a greater ratio of the data transfer time and the computing time in the cloud, and thus results in a poor cost efficiency. The idea of offloading tasks with small input data size to the cloud helps to improve the cost efficiency, which is one of our consideration for designing heuristics or hybrid heuristics with high effectiveness. This is the main reason why EFTF has the best cost efficiency in all of these heuristics, because EFTF iteratively assigns the task with minimal finish time, which usually having small input data size, when making the offloading and assignment decisions in the cloud.

## Processing efficiency

Figure 5 respectively show the values of the two processing efficiency metrics when applying various task scheduling methods. As shown in the figure, these two processing efficiency metric values respectively have a similar relative performance to the corresponding SLA satisfaction metric values for these scheduling methods, as shown in Figs. 3B and 3C. This is because all of these methods have comparable performance in the makespan. Thus, our method also has the best performance in processing efficiency.

## DISCUSSION

Meta-heuristics, typified by PSO, can achieve better performance than heuristics. This achievement is mainly due to their global search ability which is implemented by the randomness and the converging to the global solution during their iteratively search. But the meta-heuristic based method must be designed carefully, otherwise, it may have worse performance than heuristics, such GA, PSO_srv and BPSO, as presented in experimental results.

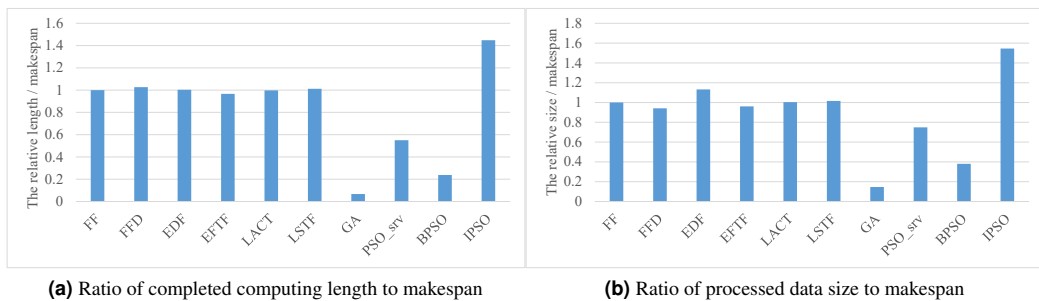

**(a)** Ratio of completed computing length to makespan      **(b)** Ratio of processed data size to makespan

**Figure 5** **The processing efficiency achieved by various scheduling methods.** (A) Ratio of completed computing length to makespan. (B) Ratio of processed data size to makespan.

The main difference between IPSO and BPSO is the search space size for a specific problem. This inspires us that more efficient encoding method with short code length helps to reduce the size of search space, and thus improve the probability of finding the global best solution.

One of the main advantages of heuristics is that they are specifically designed for targeted problems. This produces efficient local search strategies. Thus, in some times, heuristics have better performance than meta-heuristics, such as EDF vs. BPSO. Therefore, a promising research direction is integrating a local search strategy into a meta-heuristic algorithm to cover its shortage caused by its purpose of solving general problem. While, different combining of heuristic local searches and global search strategies should result in various performance improvements, which is one of our future studies.

## RELATED WORKS

As DE3C is one of the most effective ways to solve the problem of insufficient resources of smart devices and task scheduling is a promising technology to improve the resource efficiency, several researchers have focused on the design of efficient task scheduling methods in various DE3C environments (*Wang et al., 2020*).

To improve the response time of tasks, the method proposed by *Apat et al. (2019)* iteratively assigned the task with the least slack time to the edge server closest to the user. Tasks are assigned to the cloud when they cannot be finished by edges. Their work didn't consider the task scheduling on each server. OnDisc, proposed by *Han et al. (2019)*, heuristically dispatched a task to the server providing the shortest additional total weighted response time (WRT), and sees the cloud as a server, to improve overall WRT. For minimizing the deadline violation, the heuristic method proposed by *Stavrinides & Karatza (2019)* used EDF and earliest finish time first for selecting the task and the resource in each iteration, respectively. When a task's input data was not ready, the proposed method tried to fill a subsequent task before it.

Above research focused on the performance optimization for task execution, while didn't concern the cost of used resources. In general, a task requires more resources for a better performance, and thus there is a trade-off between the task performance and the resource

cost. Therefore, several works concerned the optimization of the resource cost or the profit for service providers. For example, *Chen et al. (2020)* presented a task scheduling method to optimize the profit, where the value of a task was proportional to the resource amounts and the time it took, and resources were provided in the form of VM. Their proposed method first classified tasks based on the amount of its required resources by K-means. Then, for profit maximization, their method allocated the VM class to the closest task class, and used Kuhn-Munkres method to solve the optimal matching of tasks and VMs. In their work, all VMs were seen as one VM class. This work ignored the heterogeneity between edge and cloud resources, which may lead to resource inefficiency (*Kumar et al., 2019*). *Li, Wang & Luo (2020)* proposed a hybrid method employing simulated annealing to improve artificial fish swarm algorithm for offloading decision making, and used best fit for task assignment. This work focused on media delivery applications, and thus assumed every task was formed by same-sized subtasks. This assumption limited the application scope. *Mahmud et al. (2020)* proposed a method which used edge resources first, and assigned the offloaded task to the first computing node with minimal profit merit value, where the profit merit was the profit divided by the slack time.

All of the aforementioned methods employed only edge and cloud resources for task processing, even though most of user devices have been equipped with various computing resources (*Wu et al., 2019*) which have zero transmission latency for users' data. To exploit all the advantages of the local, edge and cloud resources, some works are proposed to address the task scheduling problem for DE3C. The method presented in *Lakhan & Li (2019)* first tried several existed task order method, *e.g.,* EDF, EFTF, and LSTF, and selected the result with the best performance for task order. Then, the method used existed pair-wise decision methods, TOPSIS (*Liang & Xu, 2017*) and AHP (*Saaty, 2008*), to decide the position for each task's execution, and applied a local search method exploiting random searching for the edge/cloud. For improving the delay, the approach presented in *Miao et al. (2020)* first decided the amounts of data that is to be processed by the device and an edge/cloud computing node, assuming each task can be divided into two subtasks with any data size. Then they considered to migrate some subtasks between computing nodes to further improve the delay, for each task. The method proposed in *Zhang et al. (2019)* iteratively assigned the task required minimal resources to the nearest edge server that can satisfy all of its requirements. *Ma et al. (2022)* proposed a load balance method for improving the revenue for edge computing. The proposed method allocated the computing resources of the edge node with the most available cores and the smallest move-up energy to the new arrived task. To improve the total energy consumption for executing deep neural networks in DE3C with deadline constraints, *Chen et al. (2022)* proposed a particle swarm optimization algorithm using mutation and crossover operators for population update. *Wang et al. (in press)* leveraged reinforcement learning with sequence-to sequence neural network for improving the latency and the device energy in DE3C. Machine learning-based or metaheuristic-based approaches may achieve a better performance than heuristics, but in general, they consume hundreds to tens of thousands more time, which makes them not applicable to make online scheduling decisions.

Different from these above works, in this paper, we design an Integer PSO based hybrid heuristic method for DE3C systems. Our work is aiming at optimizing SLA satisfaction and resource efficiency, and trying to jointly address the problems of offloading decision, task assignment, and task ordering.

## CONCLUSION

In this paper, we study on the optimization of the SLA satisfaction and the resource efficiency in DE3C environments by task scheduling. We formulate the concerned optimization problem as a BNLP, and propose an integer PSO based task scheduling method to solve the problem with a reasonable time. Different from existing PSO based methods, our method exploits the integer coding of the task assignment to cores, and rationalizes the position of each particle by rounding and modulo operation to preserve the particle diversity. Simulated experiment results show that our method has better performance in both SLA satisfaction and resource efficiency compared with nine classical and recently published methods.

The main advantages of our method are the efficient encoding method and the integration of meta-heuristic and heuristic. In the future, we will continue to study on more effective encoding methods and try to design hybrid methods by hybridizing meta-heuristic and heuristic search strategy for a better performance.

## ACKNOWLEDGEMENTS

The authors would like to thank the anonymous reviewers for their valuable comments and suggestions.

### Funding

The research was supported by the Key Scientific and Technological Projects of Henan Province (Grant No. 202102210174, 212102210096, 202102210383, 212102210410, 202102210149, 212102210382, 212102210104, 212102210424, 212102210418), the Key Scientific Research Projects of Henan Higher School (Grant No. 20B520039, 21A520050), the National Natural Science Foundation of China (Grant No. 61872043, 61975187, 62072414), the Qin Xin Talents Cultivation Program, Beijing Information Science and Technology University (No. QXTCP B201904), and the fund of the Beijing Key Laboratory of Internet Culture and Digital Dissemination Research (Grant No. ICDDXN004). The funders had no role in study design, data collection and analysis, decision to publish, or preparation of the manuscript.

### Grant Disclosures

The following grant information was disclosed by the authors:
The Key Scientific and Technological Projects of Henan Province: 202102210174, 212102210096, 202102210383, 212102210410, 202102210149, 212102210382, 212102210104, 212102210424, 212102210418.

The Key Scientific Research Projects of Henan Higher School: 20B520039, 21A520050.
The National Natural Science Foundation of China: 61872043, 61975187, 62072414.
Qin Xin Talents Cultivation Program.
Beijing Information Science and Technology University: QXTCP B201904.
The Beijing Key Laboratory of Internet Culture and Digital Dissemination Research: ICDDXN004.

## Competing Interests

Junqiang Cheng is employed by Europe-Aisa Hi-tech and Digital Technology Company Limited.

## Author Contributions

- Bo Wang conceived and designed the experiments, performed the experiments, performed the computation work, authored or reviewed drafts of the paper, and approved the final draft.
- Junqiang Cheng performed the experiments, analyzed the data, prepared figures and/or tables, and approved the final draft.
- Jie Cao conceived and designed the experiments, performed the computation work, prepared figures and/or tables, and approved the final draft.
- Changhai Wang analyzed the data, authored or reviewed drafts of the paper, and approved the final draft.
- Wanwei Huang conceived and designed the experiments, authored or reviewed drafts of the paper, and approved the final draft.

## Data Deposition

The implementation of task scheduling methods in C is available in the Supplementary File.

## Supplemental Information

Supplemental information for this article can be found online at http://dx.doi.org/10.7717/peerj-cs.893#supplemental-information.

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
