# Peer review of "Integer particle swarm optimization based task scheduling for device-edge-cloud cooperative computing to improve SLA satisfaction"

_PeerJ Computer Science, doi:10.7717/peerj-cs.893_

## Round 0.1 · original submission · Major Revisions

Kindly fully address all the concerns raised by the reviewers and you may resubmit the manuscript for further evaluation.

·

Basic reporting

1. Abstract has to be rewritten. A concise and factual abstract is required.
2. Problem/gap not properly highlighted in the abstract, also quantitative results/improvement missing in the abstract.
3. There are a number of English language-related issues and need to be revised especially a lot of conjunctions used, sentences are too long and difficult to apprehend.
4. A meta-heuristic algorithm …. Which can have a better performance than heuristic methods (line 45-47), how? Need some references or proper justification.
5. According to the assumption at line 109 that tasks assigned to a core are executed based on EDF, in such a case how do you handle starvation?
6. Do you think that branch and bound technique can’t be applied for large-scale problems (line 154)?
7. The sentence “Then, their method used Kuhn-Munkres …” need to be rephrased (line 364)
8. The sentence “Different from these … “ need to be rephrased (line 393)
9. No paper is found from 2021 and the single paper you have mentioned as of 2021 is actually published in 2018.
10. Add papers from 2021
11. Figure number should be mentioned (line 339)

Experimental design

1. The numbers of subscripts were used for variables and symbols are very high, reduce the number of subscript variables and symbols as few as possible. For this, you can divide the “problem formulation” section into subsections.

2. Do you think inertia weight has a role in the PSO algorithm? Which variant of inertia weight you have used for your algorithm. Detail about inertia weight and other terms like r1, r2, c1, c2 are not given (lines 212-213).

3. Complexity of the proposed approach has been calculated but not compared with their counterparts.

4. Which encoding scheme you have used? (Figure 2)

5. “Referring to published ……” Only mentioning that it is taken from the literature is not sufficient, you need to provide proper references (lines 247-250).

Validity of the findings

1. References not mentioned for the first three approaches (FF, FFD, and EDF) used for comparisons (lines 253, 254-255, and 256-258)? You need to provide proper references.
2. How the computing time of task on the Cloud can be greater than on the Edge? (lines 330-331)

Reviewer 2 ·

Basic reporting

The authors claimed that his approach has better resource efficiency. However, the results concerning resource efficiency are not meaningful. The authors are required to obtain results concerning Average resource utilization which will be between 0 to 1 or 0 to 100 %.
All the figures are having no y-axis caption. Similarly, some of the results seem the same even their titles are different. For instance, Figure 3 (a) and Figure 3 (c).
The authors are required to add a section results and discussion that should show why their proposed approach is better than the existing and results need to be clearly described.

Experimental design

The authors claimed that his approach has better resource efficiency. However, the results concerning resource efficiency are not meaningful. The authors are required to obtain results concerning Average resource utilization which will be between 0 to 1 or 0 to 100 %.
All the figures are having no y-axis caption. Similarly, some of the results seem the same even their titles are different. For instance, Figure 3 (a) and Figure 3 (c).
The authors are required to add a section results and discussion that should show why their proposed approach is better than the existing and results need to be clearly described.

Validity of the findings

The results need to be discussed properly.

---

## Round 0.2 · Minor Revisions

Please address all the suggested changes before re-submission.

·

Basic reporting

1. Previous comments regarding the abstract have been incorporated, however, it is recommended to start your abstract from an introductory sentence (as in the previous one) instead of a sentence like “In this paper”.
2. Authors have provided quantitative results in the abstract and in the contribution section, however, the improvements shown in the different sections are different. Moreover, in the abstract, the term “x” is used while in the contribution section the symbol “%” is used, correct them or justify if it is different?
3. In the last paragraph of section 1 “introduction”, Section 6 has mentioned two times?
4. The symbol “Z” and “U” used in equations 17 and 19 have not been explained in the text.
5. In response to previous comments “Do you think inertia weight has a role in PSO algorithm? Which variant of inertia weight you have used for your algorithm. Detail about inertia weight and other terms like r1, r2, c1, c2 not given (lines 212-213)”, it is mentioned that it is out of scope of this paper. I agree up to some extent that you are not working the inertia weight strategies but Inertia weight is one of the most important control parameters for maintaining the balance between the global and local search of the PSO, which can affect the performance of the PSO. Moreover, the values of constants like c1 and c2 also matter.
Therefore, it is suggested to use the latest variant of inertia weight strategy or at least refer/cite some latest papers in the text so that readers can explore them if interested to follow the details, some of the latest papers in this regard are given below.
https://doi.org/10.1007/s11227-021-04062-2.
https://doi.org/10.1007/s10586-019-02983-5

Experimental design

N/A

Validity of the findings

N/A

Reviewer 2 ·

Basic reporting

The language used in the paper is very poor. The language of the whole paper still needs to be improved.

Experimental design

The reviewer has not incorporated changes to the article based on my first comment
"The authors claimed that his approach has better resource efficiency. However, the results
concerning resource efficiency are not meaningful. The authors are required to obtain results
concerning Average resource utilization which will be between 0 to 1 or 0 to 100 %. "

Validity of the findings

The authors have claimed almost 10 times improvement against the state-of-the-art approaches which seems incorrect. The authors are required to perform the experiments again and see whether the improvement claimed is same

---

## Round 0.3 · accepted · Accept

Congratulations, the revisions are satisfactory and the manuscript is recommended for publication.